# OpenReview forum: "Temporal Sparse Autoencoders: Leveraging the Sequential Nature of Language for Interpretability"
_ICLR.cc/2026/Conference — ICLR 2026 Oral_

### Official Review · Reviewer_z5dK · 2025-10-28

**Soundness:** 3
**Presentation:** 4
**Contribution:** 3
**Rating:** 6
**Confidence:** 4

**Summary:**

The paper introduces Temporal Sparse Autoencoders (Temporal SAEs), an extension of standard SAEs designed to handle sequential data. The key idea is to encourage features to remain active over time, capturing higher-level, temporally consistent concepts while allowing lower-level features to vary more freely. The authors motivate this through an analogy between syntactic (short-term, local) and semantic (long-term, abstract) interpretability, and propose a smoothness-based regularization term to enforce temporal coherence in activations. Experiments on language model activations show that Temporal SAEs produce more stable and interpretable latents, with some evidence of improved long-range consistency and separation between high- and low-level features.

**Strengths:**

- The distinction between syntactic and semantic interpretability, and how this relates to temporal sequences, is a novel framing. It helps clarify how different kinds of latent dynamics might reflect hierarchical structure in sequential data.

- The conceptual split between high-level, consistent latents and low-level, fast-varying ones is elegant and intuitive. It connects well to both neuroscience analogies and hierarchical representations in modern sequence models.

- Figure 1 effectively communicates the model’s core idea and its temporal behavior—it’s visually clear and supports the text well.

- The modeling assumptions are reasonable, and the paper does a good job of articulating the intuition behind them, making the method easy to follow.

- The experiment on long-range consistency adds depth, showing that the proposed method captures meaningful temporal dependencies beyond local smoothness.

**Weaknesses:**

- It’s not entirely clear what the practical takeaway is. Should one prefer semantic or syntactic features, and in what contexts? The paper motivates the distinction well, but it stops short of offering guidance on which is preferable or when. Relatedly, the utility of the proposed method remains somewhat vague. Section 4.5 shows an application of Temporal SAEs, but without a comparison to standard SAEs, it’s hard to tell whether the temporal extension is actually necessary or beneficial for that task.

- The smoothness metric feels fragile. Because it depends on the maximum activation change, it can overemphasize outliers or produce unstable values when latent differences are small. A more stable approach (such as comparing relative high-frequency energy or using multi-scale smoothness measures) would likely give a clearer and more reliable picture of temporal consistency.

- The spliced-text experiment (Figure 4) would be much stronger if it directly compared Temporal SAEs to normal SAEs. Figure 1 provides a helpful teaser, but including this comparison within the main experimental section would make the qualitative claims far more convincing.

- Table 1 leaves some gaps. It would be informative to include smoothness scores for the low-level split, since those features are expected to vary more quickly and thus might show even lower smoothness than standard SAEs. Including this would help validate the intended hierarchy between low- and high-level components.

**Questions:**

- What is the intended takeaway from the distinction between syntactic and semantic features? Should one type be preferred or emphasized in certain contexts? More broadly, what is the practical utility of Temporal SAEs compared to standard SAEs? In what kinds of tasks or analyses do the temporal extensions provide clear advantages?

- Could the authors include a direct comparison between Temporal SAEs and standard SAEs in the spliced-text experiment (Figure 4)? This would help confirm that the observed behavior genuinely depends on the temporal component rather than standard SAE properties.

- How robust is the current smoothness metric to outliers or very small latent changes? And would the authors consider alternative measures—such as frequency-domain smoothness (e.g., high-frequency energy ratio) or multi-scale smoothness—to confirm that the results are not artifacts of the chosen metric?

---

> ### Author Response · Authors · 2025-11-21
> **Author Response to Reviewer z5dK**
>
> Thank you for your comprehensive review! We are happy you appreciate our novel framing of the problem. We have responded to your questions and concerns below and have also made all suggested changes in the manuscript (shown in red).
>
> ## W1 and Q1 Part 1: Semantic vs Syntactic:
> We thank the reviewer for this suggestion, and we include a more explicit discussion of when semantic or syntactic features would be preferred in lines 320-323 of the updated manuscript. We further elaborate below:
>
> In many of the applications that SAEs are frequently used for or motivated with, such as monitoring for and auditing unsafe behavior [1], steering models [2], or understanding data at scale [3], what users are generally trying to identify and control are high-level features that correspond to relatively abstract concepts (e.g., misinformation, biases, sexually-explicit content). In these contexts, semantic features are likely preferable, as users may care less about the individual token choices and instead are trying to measure and control the overall topic of the outputs. In contrast, in applications such as coding, poetry, or measuring linguistic capabilities such as sentence parsing, users may care more about syntactic features (e.g., the next token should be a noun or a closing bracket). We note that in either case, users need not choose between the two sets of features. Rather, disentanglement between the two, as provided by T-SAEs,  allows for more precise identification and control over features.
>
>
> ## W1 and Q1 Part 2:  Practical utility of T-SAEs compared to SAEs:
> In order to further justify the practical utility of our method, we explore two additional case studies that directly compare T-SAEs to standard SAE baselines (Section 4.5 and Appendix B of updated manuscript).
>
> In the first, we use T-SAEs to explore data at scale, specifically decomposing a popular RLHF dataset (Anthropic’s HH-RLHF) to understand human preferences and surface any potential spurious correlations in the data. We find that T-SAEs are able to recover a variety of safety-relevant features while also identifying spurious correlations such as response length and compliance with user requests. Existing baselines do not recover these same concepts, often highlighting what appear to be random features.
>
> In the second case study, we evaluate whether temporal consistency improves SAE utility for steering, finding strong performance gains across a variety of features. Here, we find that T-SAEs are not only more effective at semantically changing model outputs, but they also produce more coherent outputs. Specifically, past work on steering with SAEs has found a common failure mode where for aggressive steering strengths, LLMs will begin repeating tokens. Steering involves repeatedly adding a “steering vector” at each token in a sequence during generation to control the model. As we have demonstrated, standard SAE features are highly-fluctuating, and thus consistently adding them to steer often results in shifting model latents out-of-distribution, resulting in generation failures. Furthermore, steering with syntactic features essentially overwrites only the next token generated, rather than shifting the underlying semantic content of the generation. By training high-level T-SAE features to activate consistently over a sequence, these features are more amenable to steering consistently over a sequence by changing the model’s encoding of the semantics of the task rather than simply encouraging token repetition. Please see Section 4.5 for further discussion of these case studies and the practical utility of T-SAEs.

---

> > ### Author Response · Authors · 2025-11-21
> > **Author Response to Reviewer z5dK**
> >
> > ## W2 and Q3. Smoothness metric:
> > Thank you for this feedback. We choose the Lipschitz metric because it allows us to compare two very different spaces: the sparse, nonnegative SAE feature space with an L1 norm and the more Euclidean model latent space with an L2 norm. However, you are correct that it overemphasizes outliers. To address this issue, we conduct additional smoothness experiments with your suggested frequency- and multi-scale approaches, and we also explore a wavelet-based approach to measure smoothness.
> >
> >  * Frequency-based approach: We compute an FFT of each feature and measure the ratio of high- to low-frequency power (splitting the frequency space at the midpoint).
> >  * Multi-scale approach: We take a sliding window of 8 tokens and compute a moving average filter over the sequence resulting in a smoothed signal. Then, we compute the variance of this smoothed sequence with respect to the variance of the base signal as a measure of how much signal fluctuation is low- versus high-frequency.
> >  * Wavelet-based approach: We decompose the signal into low-frequency effects represented as the average between timesteps, $(f_{t-1}+f_t)/2$ and high-frequency effects measured as the difference between timesteps, $(f_{t-1}-f_t)/2$, such that low + high = the original signal. We can iteratively (we repeat 3 times) apply this to the new average signal (smoothed signal component) while cumulatively tracking the difference (high frequency signal component). Then, we compute the variance ratio of the high to low frequency components to get a final smoothness score per feature.
> >
> > We find that, across all smoothness metrics, T-SAEs exhibit smoother high-level features. In fact, the Wavelet-based and Multiscale metrics are even more favorable to our approach, highlighting disentanglement of high-frequency information into the low-level split as you hypothesized in Weakness 4 (See discussion below). We have included this table in the Appendix (Tables 9 and 10) of our updated manuscript.
> >
> > |           | Lipz   |        |        | FFT    |        |        | Wavelet |        |         | Multiscale |          |        |
> > |-----------|--------|--------|--------|--------|--------|--------|---------|--------|---------|------------|----------------|--------|
> > |           | Total  | H      | L      | Total  | H      | L      | Total   | H      | L       | Total      | H              | L      |
> > | TSAE      | 0.12 | 0.10 | 0.15 | 0.83 | 0.75| 0.91 |  8.33 |  3.85 | 13.96 |     5.42 |         3.17 |  8.21 |
> > | MSAE      | 0.14 | 0.15 | 0.12 | 0.83 | 0.77 | 0.89 |  8.09 | 7.94 |   9.17 |     5.59 |         5.87 | 4.99 |
> > | BatchTopK | 0.13 |        |        | 0.84 |        |        |  8.99 |        |         |     6.22 |
> >
> >
> > ## W3 and Q2. Spliced Text Experiment (Figure 4):
> > Thank you for this comment! We have included 3 baselines (Matryoshka SAEs, BatchTopK SAEs, and Neuronpedia’s Matryoshka SAEs) for Figure 4 in Appendix C.2 Figure 7, showing the same qualitative trend as that of Figure 1. Due to space constraints, we are unable to move it to the main experimental section during the rebuttal period, but, space permitting, we hope to include it in the camera-ready version of the manuscript. In Figure 7, we see that baseline features often fluctuate more, and do not cleanly separate between different texts. Additionally, we find that baselines often have features firing constantly over the entire sequence regardless of semantics. In contrast, as seen in Figure 4 and 7, the T-SAE features highlight boundaries between texts and activate more smoothly within sequences.

---

> > > ### Author Response · Authors · 2025-11-21
> > > **Author Response to Reviewer z5dK**
> > >
> > > ## W4. Table 1 gaps:
> > > In the Appendix (Table 3) of our original submission, we have included smoothness metrics for the full and low-level splits for the T-SAE and our baselines as suggested by the reviewer. Indeed, your hypothesis is correct: for T-SAEs, the low-level split is actually less smooth than baselines, further providing support that T-SAEs properly disentangle high- and low-level features. Our results for the additional smoothness metrics you proposed earlier (Appendix, Tables 9 and 10) for the revised manuscript also support this claim. We appreciate your insight that this “would help validate the intended hierarchy” of our feature space, and we will move these results into the main text (Table 1) as additional support for our method, with further discussion of this behavior (lines 313-315 of updated manuscript).
> > >
> > > We sincerely thank the reviewer for your insightful comments and suggestions. We are excited to include the additional practical task evaluations, smoothness metrics, and additional baseline comparisons suggested by the reviewer in the final version of the manuscript, as we believe they improve the message of our work. If you are similarly convinced by our discussion and additional results, we would greatly appreciate it if you could consider raising your score.
> > >
> > >
> > > [1] Bricken, et al., "Towards Monosemanticity: Decomposing Language Models With Dictionary Learning", Transformer Circuits Thread, 2023.
> > > [2] Wu, Z., Arora, A., Geiger, A., Wang, Z., Huang, J., Jurafsky, D., ... & Potts, C. (2025). Axbench: Steering llms? even simple baselines outperform sparse autoencoders.
> > > [3] Movva, R., Milli, S., Min, S., & Pierson, E. (2025). What's In My Human Feedback? Learning Interpretable Descriptions of Preference Data.
> > > [4] Menon, A., Shrivastava, M., Krueger, D., & Lubana, E. S. (2025, April). Analyzing (in) abilities of saes via formal languages.
> > > [5] Park, C. F., Lee, A., Lubana, E. S., Yang, Y., Okawa, M., Nishi, K., ... & Tanaka, H. (2024). Iclr: In-context learning of representations.

---

> > > > ### Comment · Reviewer_z5dK · 2025-11-26
> > > >
> > > > Thank you for the detailed reply and new experimental results! The new steering experiments are extremely interesting! Especially the fact that increasing strengths does not push the model into OOD responses as quickly. Also, the new metrics provide more rigorous quantitative grounds for the claims that higher-level, longer temporally dependant features are captured by the T-SAE.
> > > >
> > > > Overall, I also believe these additions have made the paper much stronger. I applaud the authors for their efforts here. Additionally, I read through the other reviews and most of them are positive - the only negative feedback seems to be minor (the related work mentioned by Ahvo is not comparable) and so I will raise my score to 8.
> > > >
> > > > Small stuff
> > > >
> > > > - missing bracket in Line 460

---

### Official Review · Reviewer_e44L · 2025-10-29

**Soundness:** 4
**Presentation:** 4
**Contribution:** 3
**Rating:** 10
**Confidence:** 4

**Summary:**

The paper introduces a novel variant of the Sparse Autoencoder (SAE) architecture, called the Temporal Sparse Autoencoder (TSAE). This architecture separates features into two groups, where one group of features has a contrastive loss term encouraging similar values in adjacent tokens. This approach is theoretically motivated by considering a text to have "high-level" features like topic and tone (which should remain consistent across the text), and "low-level" features like part of speech (which varies on a token-by-token basis), and the observation that classical SAEs mostly identify the later. The paper measures TSAEs on classical SAE metrics like total variance explained (where they perform similarly) and on new metrics like ability to label MMLU question categories (where TSAEs outperform classical SAEs).

**Strengths:**

This work addresses a core shortcoming of SAEs as an interpretability technique, which is that the interpretable features they find are often too specific to individual tokens to be useful. For instance, a feature might be "Sentences endings or periods" (Figure 1), which is interpretable and useful to the SAE's reconstruction, but is not useful for downstream applications like steering.

In this regard, the addition of temporal consistency is a natural evolution of the SAE architecture. The discussion in Section 3.1 in particular is excellent at reframing our expectation of how text carries information in order to motivate the temporally-informed architecture.

The authors include clear details to facilitate reproduction of their architecture and experiments.

The authors demonstrate the advantages of their technique with a range of evidence, including quantitative experiments (Figure 4 and Table 1), a case study (Section 4.5), and visually striking diagrams (Figures 1 and 2). The case study also provides a possible application of this technique.

**Weaknesses:**

The contrastive loss has a complicated structure, and the authors do not motivate or explain what it has that form. There is insufficient explanation of why this contrastive loss outperforms the naive temporal similarity loss term (Lines 454-458 and Table 2).

The case study in Section 4.5 shows quantitative results, but does not compare them to a classical SAE. This makes it hard to judge whether the TSAE architecture is an improvement over previous methods in this context.

**Questions:**

Figure 1: What language is this copy of the Bhagavat Gita in? If it is an English translation, please indicate that, since otherwise a reader may be confused as to why the "Non-english text" feature fails to fire.

Line 202: Aren't the two terms in $L_{contr}$ the same? In the second term, if one renames $i$ to $j$ and vice-versa, and uses the symmetry of $s(x,y)=s(y,x)$, doesn't one get the first term?

Line 202: Would it be possible to test a linearized $L_{contr}$? In particular, with modest approximations it is possible to simplify $L_{contr}$ to $- \displaystyle \frac {1} {N} \sum_i s_{i,i} + \frac{1}{N^2} \sum_{i,j} s_{i,j}$, where $s_{i,j}$ denotes $s(z_t^{(i)}, z_{t-1}^{(j)})$. This would make the contrastive loss more immediately understandable, and might speed training.

To achieve this simplification, consider the log term for a single $i$ in $L_{contr}$:

$\log(\frac{e^{s_{i,i}}}{\sum_j exp(s_{i,j})}) = s_{i,i} - log(\sum_j e^{s_{i,j}})$

$≈ s_{i,i} - log(\sum_j (1+s_{i,j}))$

$= s_{i,i} - log(N + \sum_j s_{i,j}))$

$= s_{i,i} - log(N) - log(1+\frac 1 N \sum_j s_{i,j})$

$≈s_{i,i} - log(N) - \frac{1}{N} \sum_j s_{i,j}$

where the two approximations are via the taylor series approximations $exp(x) \approx 1+x$ and $log(1+x) \approx x$, respectively.

Now, discarding the $log(N)$ for being constant, averaging over the $i$, and negating, we reach

$L_{contr} ≈ - \frac 1 N \sum_i s_{i,i} + \frac{1}{N^2} \sum_{i,j} s_{i,j}$

If the cosine similarities fall in a tighter range (e.g. they are almost always in the range [0,1]), it might be appropriate to linearize around a point besides x=0.

Line 331: When you calculate AutoInterp scores, how do you handle dead latents? Are the excluded from the average for the autointerp score?

---

> ### Author Response · Authors · 2025-11-21
> **Author Response to Reviewer e44L**
>
> We thank you for your comments and feedback. We are glad you found our discussion “excellent” and our figures “visually striking.” We answer your questions and concerns below and have made your suggested edits in the updated manuscript.
>
> ## W1. Structure of the contrastive loss:
> Thank you for this question. We provide motivation and explanations as follows, with corresponding edits to the manuscript. The contrastive loss we use is the standard contrastive loss from papers such as SimCLR, CLIP, and Contrastive Predictive Coding [1,2,3]. We will provide a more thorough explanation of why it has this form and intuition for why it outperforms the naive baseline below. We have also updated the Methods section of the manuscript to incorporate this as well (lines 197-200).
>
> Our choice of contrastive loss is motivated by a large body of literature in representation learning and multimodal learning that leverages contrastive losses for unsupervised learning [1,2,3]. We use the same standard contrastive loss formulation, where our positive pairs are feature activations of tokens from the same sequence and negative pairs are activations of tokens from different sequences.
>
> Intuitively, the contrastive loss extracts more signal per batch by involving N^2 comparisons between samples. As we know that, on average, two random sequences will be semantically distinct, we can use a contrastive loss to push these distinct sequences away from each other in T-SAE feature space while simultaneously encouraging temporally-adjacent features to be close together. The reason for its improved performance over the naive temporal similarity baseline is that the naive similarity baseline only requires temporally-adjacent feature activations to be similar, while the contrastive loss additionally encourages different sequences to have different feature activations to fully use the high-level feature space. This has two additional benefits: (1) features are encouraged to spread out and (2) activation similarity cannot be achieved with a trivial collapsed solution. Because the naive loss only encourages neighboring feature activations to be more similar, it can be trivially achieved by having a constant feature activation over all data. This hurts the utility of the high-level feature space and prevents it from developing meaningful features that differentiate semantics across different sequences. In fact, even in standard SAEs, recent work [4] has found features which activate on all tokens. The contrastive loss mitigates this by encouraging feature diversity over a batch. Empirically, we see that this added signal results in a richer semantic representation in our ablation study (Table 2).
>
> ## W2. Baseline in safety case study (Section 4.5):
>
> Thank you for this suggestion. In our updated version of the manuscript, we include a baseline for the dataset understanding case study and results on an additional case study of steering with baselines (Section 4.5 and Appendix B). First, we use T-SAEs and Matryoshka SAEs to decompose Anthropic’s HH-RLHF dataset to gain a conceptual understanding of user preferences on harmful/harmless data. We find that T-SAEs, by providing a more high-level description of trends in user preferences, a better understanding of user preferences and can correctly identify spurious trends allowing for data cleaning for downstream alignment. Second, we conduct model steering with both SAEs. Here, we find that T-SAEs are not only more effective at semantically changing model outputs, but they are also more coherent. Specifically, past work on steering with SAEs has found a common failure mode where at aggressive steering strengths, the LLM will begin repeating tokens. Our analysis finds that this may be due to the syntactic nature of existing SAEs, and the semantic features found by the T-SAE don’t suffer from this failure mode, even at large steering strengths.
>
> ## Q1. Language of Bhagavat Gita:
> Thank you for catching this potential source of confusion! The text is an English translation of Sanskrit (pasted below). We will clarify this in the caption of Figure 1 and include all prompts used to generate plots in the associated codebase.
>
> > “Oh noblest of men, that person of wise judgement equipoised in happiness and distress, whom cannot be disturbed by these is certainly eligible for liberation. The impermanent has no reality; reality lies in the eternal. Those who have seen the boundary between these two have attained the end of all knowledge. But know that by whom this entire body is pervaded, is indestructible. No one is able to cause the destruction of the imperishable soul."

---

> ### Author Response · Authors · 2025-11-21
> **Author Response to Reviewer e44L**
>
> ## Q2. Contrastive Loss Terms:
> Thank you for pointing this out, there is a typo in the equation. The left hand denominator should sum over $z_{t-1}$ and the right hand denominator should sum over $z_{t}$ terms, thus they are not the same. We note that this does not affect our implementation of the loss in our code.
>
> $\mathcal{L}_{\mathrm{contr}}$
>
> $$
> = -\frac{1}{N}\sum_{i=1}^N \log\frac{e^{s(z_{t}^{(i)},\, z_{t-1}^{(i)})}}{\sum_{j=1}^N e^{s(z_{t}^{(i)},\, z_{t-1}^{(j)})}}
> -\frac{1}{N}\sum_{j=1}^N \log\frac{e^{s(z_{t}^{(j)},\, z_{t-1}^{(j)})}}{\sum_{i=1}^N e^{s(z_{t}^{(i)},\, z_{t-1}^{(j)})}}
> $$
>
> We have updated the manuscript to correct this equation. Put another way you can view the first term as contrasting the $z^i_t$th term with terms $z^j_{t-1}$, for all $j$ such that $j \neq i$ and the second term as contrasting the $z^j_{t-1}$st term with terms $z^i_t$ for all $i$ such that $i \neq j$, where $i,j$ are batch indices. This means the LHS compares one $t$ term with all $t-1$ terms, whereas the RHS compares one $t-1$ term with all $t$ terms. To symmetrize the contrastive loss, it is necessary to have both terms.
>
> ## Q3. Testing a linearized contrastive loss:
> The linearization you propose is interesting and intuitive, and it may also improve computational efficiency. Unfortunately, due to time and computational constraints during the rebuttal period, we are unable to fully train and evaluate models with this modification to the loss. We note that the equation presented in the paper is simply computing the Binary Cross-Entropy loss over cosine similarities for paired and unpaired samples. The BCE loss is in general more robust and scale invariant, likely explaining its adoption in past literature. However, we believe this linearized approach is particularly relevant because we compute cosine similarity in the SAE activation space (which is sparse and non-negative), which means we know that cosine similarity is in the range of [0,1] and often near 0. In fact, while cross entropy might be the right approach for normal euclidean representation space (where we want to encourage negative cosine similarity between unpaired samples), this linear loss may be better suited for the sparse, nonnegative SAE feature space. We have included a discussion on this point in the Future Work section (Appendix A.2).
>
> ## Q4. Handling of dead latents in AutoInterp scores:
> We calculate AutoInterp scores using the open-source library, SAEBench [5]. The repository does not generate an explanation or score for dead latents, so they are not included in the computation of the average. While an alternative formulation may instead penalize dead latents, we use the implementation from SAEBench to allow for maximum comparability with existing benchmarks and prior literature.
>
> We thank you for your detailed commentary and feedback! We appreciate the suggestions you have provided, which have improved our submission. We hope we have sufficiently addressed your comments. Please let us know if you have any additional questions and we will be happy to discuss further.
>
> [1] Chen, T., Kornblith, S., Norouzi, M., & Hinton, G. (2020, November). A simple framework for contrastive learning of visual representations.
> [2] Oord, A. V. D., Li, Y., & Vinyals, O. (2018). Representation learning with contrastive predictive coding.
> [3] Radford, A., Kim, J. W., Hallacy, C., Ramesh, A., Goh, G., Agarwal, S., ... & Sutskever, I. (2021, July). Learning transferable visual models from natural language supervision.
> [4] Papadimitriou, I., Su, H., Fel, T., Kakade, S., & Gil, S. (2025). Interpreting the linear structure of vision-language model embedding spaces.
> [5]: Karvonen, A., Rager, C., Lin, J., Tigges, C., Bloom, J., Chanin, D., ... & Nanda, N. (2025). Saebench: A comprehensive benchmark for sparse autoencoders in language model interpretability

---

### Official Review · Reviewer_Ahvo · 2025-10-30

**Soundness:** 3
**Presentation:** 4
**Contribution:** 2
**Rating:** 6
**Confidence:** 4

**Summary:**

The paper targets a limitation in standard SAEs used for interpretability: by treating tokens as i.i.d, they disproportionately discover local, token-specific, and often syntactic features  failing to capture higher-level semantic concepts. The authors sensibly argue this is a mismatch with the structure of language, where semantics evolve smoothly while syntax is more local. They propose Temporal Sparse Autoencoders which modify the SAE framework by (a) partitioning the feature dictionary into a "high-level" (semantic) set and a "low-level" (syntactic) set, using a Matryoshka-style residual loss.It introduces a novel temporal contrastive loss that encourages the high-level features to have similar activations for adjacent tokens, while pushing them to be dissimilar from other tokens in the batch. Experiments on Pythia-160m and Gemma-2-2b show that this simple addition works quite well. Qualitative results on spliced text show T-SAEs identify smooth, semantic features that cleanly track topic shifts, unlike the noisy, high-frequency activations of baseline SAEs (like matryoshka SAE). Probing confirms T-SAE high-level features are far better at capturing semantic and contextual information, while the low-level features capture syntax. The authors also noted that improvement in feature quality is achieved without sacrificing standard reconstruction performance.

**Strengths:**

The presentation of the paper is very clear: the motivation, method, and results are all presented in a way that is easy to follow. The spliced-text visualizations are particularly strong, providing an "it just works" demonstration that is more compelling than the quantitative metrics alone.

The experimental validation is robust. The authors demonstrate their method's contribution through: (a) The smooth, semantic features in the visualizations are neat, (b) probing results confirm the high-level/low-level disentanglementm, and (c) the method maintains competitive performance on standard SAE benchmarks, showing the gains aren't at the cost of reconstruction fidelity.

Despite some novelty concern (detailed below), the results are solid. It confirms that even simple temporal priors can significantly improve the usefulness of SAE features, moving the analysis from token-level only to be useful on a sequence-level as well.

**Weaknesses:**

A major weakness of this work is the overall lack of proper contextualization of their work against highly relevant works. The central problem—that the i.i.d. assumption for tokens is a flaw and corresponding solution that temporal dynamics should be leveraged for smoother extracted features is not new. For instance, this paper published earlier at iclr 2025 has identified very same problem and proposed a similar temporal modification to SAE (https://scholar.google.com/citations?view_op=view_citation&hl=en&user=ycscpaQAAAAJ&citation_for_view=ycscpaQAAAAJ:2osOgNQ5qMEC).

Similarly, as an important and related problem (as the authors admit in related works), I believe feature absorption deserves more discussion. Even if not adding some experiments on feature absorption, which should be straightforward given the authors already use SAEBench (https://github.com/adamkarvonen/SAEBench), at least discuss why and contextualize with previous SOTA/prior works on this benchmark, such as https://openreview.net/forum?id=i31cKXiyim and https://www.alignmentforum.org/posts/zbebxYCqsryPALh8C/matryoshka-sparse-autoencoders (both are concurrent with the Matryoshka SAE that the authors cited but still relevant).

Another area that the model can improve upon is the limited scale. With experiments mostly focused on Pythia-160m, and Gemma-2-2b, it is plausible, although eventually unclear, whether the results will generalize mech interp as a field is moving towards larger model.

**Questions:**

On the feature split: The 20%-80% split between high- and low-level features seems rather arbitrary. How sensitive is the model to this split? And more importantly, are there good rules of thumb when setting this value for a new model/layer without running a sweep?

On the adjacent token assumption: The contrastive loss relies on $x_t$ and $x_{t-1}$. This is a very local definition of semantic smoothness. Have you explored other sampling strategies, such as a small window of recent tokens or a distance-weighted contrastive loss?

My rating is currently marginal towards accept/reject, although I'm willing to raise my rating subject to the weaknesses/questions being sufficiently addressed.

---

> ### Author Response · Authors · 2025-11-21
> **Author Response to Reviewer Ahvo**
>
> We thank you for your comments and feedback. We are glad you found our figures compelling and appreciated your comment that “it just works.” We answer your questions and concerns below and have made your suggested edits in the updated manuscript.
>
> ## W1. Contextualization against Related Works:
> Thank you for pointing us to this relevant work! We will be sure to discuss [1] in the camera-ready version. We provide an overview of how our work differs in motivation, method, and evaluation below. Finally, we would also like to highlight to the reviewer that, to the best of our knowledge, ***[1] was only made publicly available on October 9th, after the submission deadline for ICLR 2026***, thus we were unable to cite it in our submission.
>
> Importantly, this work differs from ours in a few key ways:
> **Motivation:** [1] is focused on resolving the problem of feature absorption, an important challenge facing SAEs in the literature. In contrast, our work is focused on leveraging properties of natural language to improve the recovery of semantic content in SAE decompositions. Our main motivation is to improve the semantic structure by disentangling semantics and syntax, and therefore improve the interpretability and downstream applicability of the feature space. While complimentary, we find these motivations are distinct.
> **Method:** [1] uses a mask to stabilize important features during training, where the mask is a probabilistic threshold of importance scores based on exponential moving averages over time. The goal of their approach is to track feature importance, whereas our loss function is designed to make a subset of features that are high-level smooth over time using a temporal contrastive loss. From this perspective, [1] leverages the temporal smoothness assumption to optimize for absorption whereas we directly optimize for smoothness in a subset of high-level features, and enforce disentanglement of the remaining low-level features with a nested Matryoshka structure.
> **Evaluation:** As the goal of [1] is to resolve feature absorption, the authors benchmark their method with standard feature absorption evaluations. In contrast, we are mainly interested in how temporal consistency improves downstream utility of SAEs, particularly in our ability to understand how semantics changes over time, how semantics and syntax are disentangled in LLMs, and how semantics can be leveraged for dataset understand and model steering.
>
> In the paper, we will add the citation in related work (Lines 124-126):
>
> > Finally, concurrent work [1] also explores a notion of temporal consistency, using an exponential moving average over time to address feature absorption.
>
> ## W2. Feature Absorption:
> Thank you for this comment. We are happy to provide experimental results on feature absorption as suggested by the reviewer; however, we note that mitigating absorption is not the primary goal of this work. We also note that T-SAEs already incorporate the SOTA architecture the reviewer suggested (MatryoshkaBatchTopK) which has been shown to already address absorption. On lines 116-118 of the original submission, we address the problem of feature absorption and how state-of-the-art methods such as [4] were proposed to resolve it. We have modified this discussion (lines 117 of the updated manuscript) to include the additional reference [2] provided by the reviewer.
>
> To study the level of absorption in T-SAEs, we ran the SAEBench absorption evaluation on our T-SAE models and Matryoshka baselines and found that they have quite comparable feature absorption scores, and T-SAEs have fewer split features. We will include this table in the Appendix of the revised manuscript (Appendix C.3).
>
> |                | Mean Absorption Frac. Score | Mean Absorption Full Score | Num Split Feature |
> |----------------|-----------------------------|----------------------------|-------------------|
> | Temporal SAE   | 0.24 +/- 0.16               | 0.28 +/- 0.16              |1.96 +/- 1.18      |
> | Matryoshka SAE | 0.22 +/- 0.16               | 0.26 +/- 0.17              |   2.19 +/- 1.74   |

---

> ### Author Response · Authors · 2025-11-21
> **Author Response to Reviewer Ahvo**
>
> ## W3. Choice of Model:
> Thank you for this suggestion. We focus on Pythia-160m and Gemma-2-2b to allow for comparability with publicly available SAE baselines. We further highlight that these models and sizes are commonly used in prior literature [3-8] and note this in our updated manuscript on line 257. We believe that the results on these models will generalize to larger models, as the modification we make is scale invariant and does not rely on a different architecture or more data (our temporal contrastive loss simply uses existing data more effectively).
> To further verify the scalability of our approach, we have trained a T-SAE on Llama-3.1-8b-Instruct for a short period of time (20% of the number of gradient steps for Pythia/Gemma) and observe similar trends in terms of train FVE, reconstruction loss, temporal loss, and dead features. We additionally include TSNE results as proof of concept for the Llama T-SAE in Appendix C.4. Due to computational constraints, we are unable to fully train and evaluate on large models during the rebuttal period.
>
> ## Q1. 20-80 Split Choice:
> In our original submission, we performed an ablation study to test the sensitivity to this exact hyperparameter (Table 2). In particular, we test feature splits such as 10-90 and 50-50 and discuss their implications on lines 444-447 of the original submission. We observe that our method is not particularly sensitive to this hyperparameter, but we do see evidence that significantly increasing the size of the high-level split improves semantic and contextual recovery at the expense of syntactical information. This points to a more general rule of thumb: depending on the downstream SAE use case, if the practitioner cares more about semantic recovery then a more balanced split will be more useful. Alternatively, if syntactic recovery is more important, then a practitioner may prefer to have fewer high-level features. We do not believe a fine-grained sweep is necessary to set the value for this split, and larger adjustments can be made with this broader intuition. Furthermore, temporal loss, smoothness, and FVE can all be tracked during training to provide effective and quick feedback on the hyperparameter choice.
>
> ## Q2. Adjacent token assumption:
> We agree that these are indeed intuitive alternatives to our proposed setup and we point the reviewer to our discussion of ablation studies in the original submission (Table 2), which test exactly what the reviewer suggests: a less local definition of semantic smoothness given by an alternative sampling strategy to the direct previous token. Specifically, we explore sampling a token at random from the preceding tokens (t-rth token with r < t) rather than the t-1st token for our contrastive pair, and we also explore a noncontrastive version of the loss and discuss these choices on lines 447-454 of the original submission. We find that sampling a random token from the past improves recovery of contextual information without improving semantic recovery. This intuitively makes sense as we are providing more global context to the SAE during training, but local information already sufficiently captures semantics. In our initial exploration of the problem, we tested the alternative suggestion provided by the reviewer of a distance-weighted loss; however we found that a contrastive formulation was superior due to its leveraging of negative pairs, and the contrastive loss and distance weighting could not be combined because contrastive losses rely on binary cross entropy.
>
> We appreciate the suggestions provided by the reviewer, and we hope we have adequately addressed all of the questions and concerns. If so, we kindly ask you to consider raising your score. Please let us know if any questions remain, and we are happy to discuss further.
>
> [1] Li, T., & Ren, J. (2025). Time-Aware Feature Selection: Adaptive Temporal Masking for Stable Sparse Autoencoder Training. Workshop on XAI4Science, ICLR, 2025.
> [2] Li, E., & Ren, J. (2025). Unlocking hierarchical concept discovery in language models through geometric regularization, Workshop on BuildingTrust, ICLR, 2025.
> [3] Bussmann, B., Nabeshima, N., Karvonen, A., & Nanda, N. (2025). Learning multi-level features with matryoshka sparse autoencoders. ArXiv.
> [4]: Paulo, G., & Belrose, N. (2025). Sparse autoencoders trained on the same data learn different features.  ArXiv.
> [5]: Bussmann, B., Leask, P., & Nanda, N. (2024). Batchtopk sparse autoencoders.  ArXiv.
> [6]: Dunefsky, J., Chlenski, P., & Nanda, N. (2024). Transcoders find interpretable llm feature circuits. NeurIPS, 2024.
> [7]: Paulo, G., & Belrose, N. (2025). Sparse autoencoders trained on the same data learn different features.  ArXiv.
> [8]: Marks, S., Rager, C., Michaud, E. J., Belinkov, Y., Bau, D., & Mueller, A. (2024). Sparse feature circuits: Discovering and editing interpretable causal graphs in language models. ArXiv.

---

> > ### Comment · Reviewer_Ahvo · 2025-11-26
> >
> > The authors sufficiently addressed my concerns regarding the novelty of the work and have provided data on absorption, which I thought is a useful benchmark to measure against. For these reasons, I raised the contribution score from 2 (fair) to 3 (good) and the overall score to 8.

---

### Official Review · Reviewer_2LJx · 2025-10-30

**Soundness:** 2
**Presentation:** 2
**Contribution:** 3
**Rating:** 4
**Confidence:** 2

**Summary:**

In this paper, the authors argues that standard SAEs have some problems, e.g., recover mostly shallow, token-local, often syntactic features because they ignore a basic property of language. To address this issue, the authors propose Temporal SAE (T-SAEs), which add a contrastive-style temporal consistency loss that encourages high-level features to stay active across adjacent tokens in a sequence. For experiments, the authors try multiple models and datasets. Results show that the proposed  T-SAEs uncover richer semantic structure than vanilla SAEs.  The authors  also show these semantically cleaner features can help real tasks such as surfacing safety-relevant behaviors. Overall, the topic is interesting, but there are certain concerns that need to be addressed.

**Strengths:**

1.The topic about temporal SAE is promising and interesting.

2. The experiments are conducted on multiple models and datasets.

3.The visualizations of tnse are impressive.

**Weaknesses:**

[1] In Figure 1 (c, what is the number in x-axis mean? Does it mean the location in a long sentence?

[2] Could you also discuss the limitations of this study, and potential future directions?

[3] the experiments are conducted on models Pythia-160m and Gemma2-2b, with small parameter sizes. The reviewer understands this might be constrained by computational resources. I am not asking for additional experiments. However, could you discuss the motivations for choosing these models, and whether the results observed on these models will generalize to bigger models?

[4] in Line 252, the authors mentioned the training is on layer 8 and layer 12. COuld you explain why other layers are not selected?


[5] The writing could be further polished. For example, in the Introduction, more references should be added, e.g., reference to SAE in line 34.

[6] In the second paragraph of Intriduction, could you provide some empirical results or references to support the claim in the first sentence?

[7] In line 77 and 78, authors mention that “This feature should remain active throughout the entire sentence, because the semantic content does not change from word to word” . However, it could be possible, the content change suddenly in a sentence or paragraph. For example, we could first talk about the whether, and then talk about our cats or dogs.

[8] Could you explain more about Figure 2? This figure seems comprehensive, but missing detailed explanations.

[9] In section 4.3, are these metrices proposed by this paper or adopted from existing studies?

**Questions:**

[1] In Figure 1 (c, what is the number in x-axis mean? Does it mean the location in a long sentence?

[2] Could you also discuss the limitations of this study, and potential future directions?

[3] the experiments are conducted on models Pythia-160m and Gemma2-2b, with small parameter sizes. The reviewer understands this might be constrained by computational resources. I am not asking for additional experiments. However, could you discuss the motivations for choosing these models, and whether the results observed on these models will generalize to bigger models?

[4] in Line 252, the authors mentioned the training is on layer 8 and layer 12. COuld you explain why other layers are not selected?


[5] The writing could be further polished. For example, in the Introduction, more references should be added, e.g., reference to SAE in line 34.

[6] In the second paragraph of Intriduction, could you provide some empirical results or references to support the claim in the first sentence?

[7] In line 77 and 78, authors mention that “This feature should remain active throughout the entire sentence, because the semantic content does not change from word to word” . However, it could be possible, the content change suddenly in a sentence or paragraph. For example, we could first talk about the whether, and then talk about our cats or dogs.

[8] Could you explain more about Figure 2? This figure seems comprehensive, but missing detailed explanations.

[9] In section 4.3, are these metrices proposed by this paper or adopted from existing studies?

---

> ### Author Response · Authors · 2025-11-21
> **Author Response to Reviewer 2LJx**
>
> We thank the reviewer for your comments and feedback. We are happy you found the topic interesting and promising, and we address your questions and concerns below.
>
> ## W1/Q1. Fig. 1 x-axis units:
>
> In Figure 1, we plot SAE decompositions over a sequence of text, where the x-axis indicates each token index in the sequence. Indeed, this can be thought of as the temporal location of each token in a sentence. At each token we compute an individual SAE decomposition resulting in a [num_tokens x num_sae_features] matrix. We then gather the feature activations for the top-k most active features, resulting in k lines of length num_tokens which we plot in Figure 1. We have further clarified this in the axis labels for Figure 1 of the updated manuscript.
>
> ## W2/Q2. Limitations and Future Directions:
>
> We are happy to provide an expanded discussion of limitations and potential future directions. We have added this text to the Appendix A due to space constraints.
>
> **Limitations.** In order to compute the temporal contrastive loss term, each training data batch must include a second batch of tokens corresponding to the previous token’s activations, requiring smaller batch sizes for the same memory budget. Additionally, in this work we only explored splitting the feature space once into a single high-level and low-level feature space, but prior literature in sparse coding has shown that incorporating multiple scales of features can help to further disentangle sparse hierarchical signals and allow for more granularity on the temporal scale [1]. However, increasing the number of splits would further increase memory and computational complexity costs.
> **Future Directions.** Building off of the above, one promising future direction is to explore multiple temporal hierarchies, which, for example, could correspond to book-, paragraph-, sentence-, and token-level information. By including more hierarchies, we expect to see even more disentanglement of semantic features at different scales. Given that T-SAEs allow for consistent tracking of features over sequences, another promising future direction may be to use the features learned by SAEs as “state” trackers, allowing for easy detection of significant changes in model behavior.
>
> ## W3/Q3. Motivations for Pythia-160m and Gemma-2-2b:
> Thank you for this question. We choose these specific models due to their standardized implementation, benchmarking on SAEBench, and public availability on Neuronpedia and [SAELens](https://decoderesearch.github.io/SAELens/), allowing us to validate that our trained baselines match the performance of prior literature, such as the [Matryoshka SAE](https://www.neuronpedia.org/res-matryoshka-dc). We further note that these specific models and model sizes are standard in interpretability and specifically SAE literature [1,2,3,4] and note this in the updated manuscript (lines 257). We believe that the results on these models will generalize to larger models, as the modification we make is scale invariant and does not rely on a different architecture or more data (our temporal contrastive loss simply uses existing data more effectively).
>
> To further verify the scalability of our approach, we trained a T-SAE on Llama-3.1-8b-Instruct for a small number of steps (20%) and observe similar trends in terms of train FVE and loss. We additionally include tSNE results as proof of concept for the Llama T-SAE in Appendix C.4.
>
> ## W4/Q4. Choice of Layers for training SAEs:
> We train on these specific layers to allow for easy comparison with prior literature and publicly available benchmarks, as we noted in the Hyperparameters section, lines 254-255, of the original manuscript. These layers are currently the standard for Pythia and Gemma, as seen in [1-4].
>
> ## W5/Q5. Writing:
> Thank you for raising this comment. We will polish the writing of the introduction. Specifically, we will include additional citations as suggested. We have included excerpts of our citations below and also include them in the updated manuscript.
>
> > that humans can understand (Kim et al. (2018)), evaluate (Koh et al., 2020), and ultimately control (Wu et al., 2025).
>
> > Recent dictionary learning methods, such as Sparse Autoencoders (SAEs), have shown promise in explaining language models (Bricken et al., 2023).
>
> > rather than coherent, high-level semantic concepts (Menon et al., 2025; Paulo & Belrose, 2025)
>
> > Human language is inherently structured: meaning is conveyed through context and semantics that evolve smoothly over time, while syntax is governed by more local dependencies (Chomsky, 1965; Griffiths et al., 2004).

---

> ### Author Response · Authors · 2025-11-21
> **Author Response to Reviewer 2LJx**
>
> ## W6/Q6. Empirical evidence for the claim of the first sentence:
>
> We added more empirical evidence and references for this claim. Specifically, we will include citations to [8-10] (line 42-43 of updated manuscript), which all note the failures of existing feature explanations to provide semantically-relevant descriptions and their implications for downstream tasks. We will also provide further evidence beyond Footnote 1 and Figure 1, which respectively highlight the highly local, syntactic nature of concepts learned by current SOTA SAEs and the noisiness of decompositions. In particular, we have provided a non-exhaustive list of 60+ features from Neuronpedia’s Matryoshka SAE of layer 12 of Gemma2-2b, all of which correspond to variations of “the definitive article ‘the’” in Appendix D. We have amended our footnote to say the following: “Neuronpedia, Feature 11795 of Gemmascope’s Gemma2-2b, Residual, 16k SAE, see Appendix D for more examples”. We also provide another example of sequence decomposition with existing SAE baselines by expanding Figure 4 to include baselines in Figure 7, Appendix C.2
>
> ## W7/Q7. Context changing over a sentence or paragraph:
> Thank you for highlighting this important point. We clarify our point below and further explain the linguistic motivation and justification for our claim as follows.
>
> The field of linguistics often categorizes language into five main components: Pragmatics, semantics, phonology, morphology, and syntax. Whereas phonology, morphology, and syntax are governed by units at or smaller than the word- or token-level, semantics and pragmatics correspond to sentence meaning and language context. Thus, they do not generally vary significantly from word to word. That said, we agree that semantics can still change significantly over the course of a sequence, and we do not restrict our model to require semantics to be strictly constant over time. Prior work in computational linguistics has argued that semantics exhibit long-range dependencies, on the order of sentences or even full documents [11]. Building off of this, we propose that high-level features evolve smoothly over time, motivating our temporal consistency loss between token t and t-1. We note that we do not claim that high-level features are constant, but merely smooth in aggregate. Thus, as in your example, if semantics do change abruptly, T-SAEs can accommodate these drastic shifts, while still modelling language smoothly. Figures 1 and 4 highlight this explicitly, wherein we create sudden changes, similar to the “weather” and “cats and dogs” example you provided, by splicing together sequences from different documents. T-SAEs show clear event boundaries between spliced sequences, while exhibiting smoother high-level feature activations within each sequence.
>
> We will modify the wording of lines 77-78 (also lines 77-78 of updated manuscript) to further clarify this, and we thank the reviewer for pointing out this potential source of confusion.
>
> Old:
> > “This feature should remain active throughout the entire sentence, because the semantic content does not change from word to word”.
>
> New:
> > “We expect this feature to be active over the course of the sequence rather than simply on a few specific tokens.”

---

> > ### Author Response · Authors · 2025-11-21
> > **Author Response to Reviewer 2LJx**
> >
> > ## W8/Q8. Explanation of Fig. 2:
> > Due to space limitations, we are unable to expand the discussion of Figure 2 in the manuscript beyond what is already included in lines 285-305, but we have provided more details below. If the reviewer feels this additional information is necessary in the main paper, we are happy to include these details in the Appendix.
> >
> > In Figure 2, we qualitatively evaluate the ability of T-SAEs to recover semantic, contextual, and syntactic information.
> >
> > To evaluate semantics, we consider three datasets with topic labels corresponding to the semantic content for each sample. For example, [MMLU](https://huggingface.co/datasets/cais/mmlu) has labels of the specific subject that a question comes from, such as “Professional Law”, and [finefineweb](https://huggingface.co/datasets/m-a-p/FineFineWeb) has domain labels for each sample such as “aerospace”.
> >
> > We embed these datasets in each SAEs feature space and use TSNE to visualize the representations in two dimensions, where each point corresponds to a token from a sequence in the dataset. Each row contains the same data points and corresponds to the TSNE of the high-level T-SAE, low-level T-SAE, and Matryoshka feature spaces.
> >
> > Then, for each column, we color the data in 3 different ways corresponding to semantics, context, and syntax. For semantics, we label the data according to the semantic labels described above. For context, we label the data according to the sequence it came from. Finally, for syntax, we label the data according to its part of speech using the spaCy NLP library.
> >
> >
> > When colored by semantics (first column), we find that the high-level T-SAE feature space clusters sequences semantically, exhibiting greater structure than its low-level split and the Matryoshka baseline. Importantly, T-SAEs are not trained with semantic supervision, but are rather trained to match contexts, yet still exhibit semantic structure. For context (second column), we again see the high-level feature space better clustering data according to context than the low-level split and the Matryoshka baseline, as directly encouraged by our temporal contrastive loss. Finally, when colored by syntax (third column), we find that the low-level T-SAE feature space exhibits syntactic clustering, as does the Matryoshka Baseline.
> >
> > ## W9/Q9. Section 4.3 Metrics:
> > These metrics are commonly-used metrics to evaluate SAEs in the literature [2-6]. In particular they are a core part of SAEBench [7] which is currently the standard benchmark for SAE evaluation.
> >
> >
> >
> > We hope that our responses resolve your concerns, and that you will consider raising your score accordingly. If you have any remaining questions, we will be happy to discuss them further.
> >
> > [1]: Ong, F., & Lustig, M. (2016). Beyond low rank+ sparse: Multiscale low rank matrix decomposition.
> > [2]: Bussmann, B., Nabeshima, N., Karvonen, A., & Nanda, N. (2025). Learning multi-level features with matryoshka sparse autoencoders.
> > [3]: Paulo, G., & Belrose, N. (2025). Sparse autoencoders trained on the same data learn different features.
> > [4]: Bussmann, B., Leask, P., & Nanda, N. (2024). Batchtopk sparse autoencoders.
> > [5]: Marks, S., Rager, C., Michaud, E. J., Belinkov, Y., Bau, D., & Mueller, A. (2024). Sparse feature circuits: Discovering and editing interpretable causal graphs in language models.
> > [6]: Dunefsky, J., Chlenski, P., & Nanda, N. (2024). Transcoders find interpretable llm feature circuits.
> > [7]: Karvonen, A., Rager, C., Lin, J., Tigges, C., Bloom, J., Chanin, D., ... & Nanda, N. (2025). Saebench: A comprehensive benchmark for sparse autoencoders in language model interpretability.
> > [8] Wu, Z., Arora, A., Geiger, A., Wang, Z., Huang, J., Jurafsky, D., ... & Potts, C. (2025). Axbench: Steering llms? even simple baselines outperform sparse autoencoders.
> > [9]: Menon, A., Shrivastava, M., Krueger, D., & Lubana, E. S. (2025, April). Analyzing (in) abilities of saes via formal languages.
> > [10]: Paulo, G., & Belrose, N. (2025). Sparse autoencoders trained on the same data learn different features.
> > [11]: Griffiths, T., Steyvers, M., Blei, D., & Tenenbaum, J. (2004). Integrating topics and syntax.

---

> > > ### Comment · Reviewer_2LJx · 2025-11-27
> > >
> > > The reviewer appreciates all the hard work (e.g., new experimental results, new revisions in limitations and future work) from the authors during the rebuttal! My concerns have been addressed mostly. As a result, I will raise my score. Good work!

---

### Public Comment · ~Shijian_Xu1 · 2025-11-24
**Mistakes in the loss function**

Dear authors,

in the reconstruction losses $\mathcal L_H$ and $\mathcal L_L$, shouldn't the bias terms be subtracted, instead of added?

Also, the bias term $b^{enc}$ should be in $\mathbb R^m$ instead of $\mathbb R^d$.

Please correct me if I get it wrongly.

---

### Author Response · Authors · 2025-12-02
**Rebuttal Summary for the AC**

Dear Area Chair,

Thank you for volunteering your time and energy to chair our paper and for your service to the ICLR community. We understand that your chairing load has increased, so we provide a summary of our reviews and rebuttal to aid in this process. We are glad that all reviewers recommended acceptance of our paper after discussion (with scores of 10,8,8,6 after rebuttals from initial scores of 10,6,6,4). Further, we appreciate the reviewers' questions and suggestions as we feel that they have improved our work.

In particular, **reviewers 2LJx, Ahvo, and z5dK all agreed to raise their scores (as noted in their final discussion comments)**. Reviewers 2LJx and Ahvo both noted that their “concerns have been addressed” and reviewer z5dK found our additional experiments “extremely interesting” and “rigorous,” with our rebuttal having “made the paper much stronger.” Reviewer z5dK also commented on the reviews from the other reviewers, stating that “most of them are positive - the only negative feedback seems to be minor”. Reviewer e44L, who **already gave us a score of 10**, did not respond to our rebuttal.

## Summary of individual reviewer discussions:


### Reviewer 2LJx:
**Questions and concerns:** The reviewer mainly asked clarifying questions about figures, hyperparameter choices, and generalizability to model scaling.
**Our rebuttal:** We presented additional experimental results on a larger model, Llama 3.1-8B-Instruct, provided further details on our figures, and gave references for our specific hyperparameter choices and sweeps.
**Reviewer update post-rebuttal:** In response to our rebuttal, the reviewer noted that their “concerns have been addressed” and that they would **“raise [their] score”** accordingly. They also commented that they “appreciat[ed] all the hard work (e.g., new experimental results, new revisions in limitations and future work) from the authors during the rebuttal!”


### Reviewer Ahvo:
**Questions and concerns:** The reviewer asked about scaling to larger models, results on feature absorption, and the relation to two recent/concurrent works.
**Our rebuttal:** We presented additional experimental results on a larger model, Llama-3.1-8B-Instruct, and included a discussion of the two suggested related works. We highlight that one of these works was made publicly available after the ICLR submission deadline, and that another reviewer (z5dK) noted that these works were “not comparable.” We also pointed the reviewer to our existing ablation studies in the main text, which answered many of their questions about design and hyperparameter choice.
**Reviewer update post-rebuttal:** After our rebuttal, the reviewer responded saying that we “sufficiently addressed [their] concerns” and that they would **raise “the contribution score from 2 (fair) to 3 (good) and the overall score to 8”.**


### Reviewer z5dK:
**Questions and concerns:** The reviewer asked for a discussion of practical recommendations regarding when to use semantic or syntactic features, as well as the overall practical utility of T-SAEs. They also recommended additional smoothness metrics and comparison with baselines in Figure 4.
**Our rebuttal:** We provided a discussion of when to prefer semantic and syntactic features, and demonstrated their utility by presenting additional case studies and comparisons with baselines. We also presented results on the proposed additional smoothness metrics. All additional results supported our findings that T-SAE high-level features are smooth, that they recover disentangled features, and that T-SAEs exhibit improved utility for downstream applications like steering, data decomposition, and sequence understanding.
**Reviewer update post-rebuttal:** The reviewer's response to our rebuttal highlighted that the additional experiments and results were “extremely interesting”, “provide more rigorous quantitative grounds” for our claims, and that “these additions have made the paper much stronger.” They also commented on the reviews from the other reviewers, stating that “most of them are positive - the only negative feedback seems to be minor” and that they would **“raise [their] score to 8”.**


### Reviewer e44L:
**Questions and concerns:** The reviewer requested that we include a comparison with baseline SAEs in our case study and also asked some clarifying questions about the contrastive loss.
**Our rebuttal:** We included additional results on baseline SAE performance for our case studies, as well as results on a new downstream application of steering. Our additional experiments further confirm the utility of T-SAEs claimed in the main paper. We additionally clarified the form and purpose of the contrastive loss.
The reviewer did not respond after our rebuttal, having **already given us a score of 10.**

---

### Meta-Review · Area_Chair_QPd8 · 2026-01-05

**Summary:**

Initial reviews praised the clarity, good flow and illustrative diagrams used on the presentation of the proposed method and results of its evaluation.

The integration of temporal characteristics to SAE is a significant evolution of SAEs and was deemed to have significant impact in the field.

The evaluation of the proposed method was well received as well as the level of detail that was provided which ease reproducibility of the proposed method and reported results.

Several concerns and questions were put forward by the four reviewers to assessed this manuscript. The reviews were addressed in a meticulous manner by the rebuttal resulting in all the reviewers increasing their initial scores.

Overall, this is a very good contribution to the field and encourage its inclusion at ICLR’26.

**Reviewer Concerns:**

Addressed Concerns:

- Reviewer 2LJ

    - Missing discussions on limitations

    - Missing motivation for the models selected for the evaluation and discussion on whether the reported results would generalize to bigger models.

    - Missing motivation for the layers considered in the evaluation.

    - Writing could be polished

    - Unsupported statements at the introduction section.

    - Uncertainty on the validity of the feature consistency assumption made in lines 77-78.

    - Figure 2 is not clear

    - Source of the used metrics (Sec. 4.3).

- Reviewer Ahvo

    - Weak positioning w.r.t. highly-relevant works.

    - Missing discussion/experiments on feature absorption.

    - Experiments limited to small datasets and no signs of scalability to larger models.

- Reviewer e44L

    - Unmotivated design choices.

    - No empirical comparison w.r.t. SAEs.

    - Clarifications on contrastive loss and handling of dead latents.

- Reviewer z5dK

    - Unclear practical utility.

    - Missing empirical comparison w.r.t. SAEs.

    - Fragility of the smoothness metric.

    - Missing comparison of High and low-level components based on the smoothness metric.

    - Robustness of the smoothness metric to outliers and/or very small latent changes.


Outstanding Concerns:

- Reviewer 2LJ

    - All concerns were addressed; the reviewer increased his/her initial score.

- Reviewer Ahvo

    - All concerns were addressed; the reviewer increased his/her initial score.

- Reviewer e44L

    - All concerns were addressed; the reviewer already gave the maximum value (10) as initial score.

- Reviewer z5dK

    - All concerns were addressed; the reviewer increased his/her initial score.

**Reviewer Scores:**

The rebuttal properly addressed the concerns raised by the reviewers. Sign of this is the fact that all the reviewers were very positive with the provided rebuttal and manifested intent to increase their initial scores. Considering the actions taken by the authors and the quality of the provided responses to the concerns of the reviewers, I am convinced that additional time for discussion would had led to even higher increase of scores.

---

### Decision · Program_Chairs · 2026-01-26

Accept (Oral)